# The Complex Role Played by the Default Mode Network during Sexual Stimulation: A Cluster-Based fMRI Meta-Analysis

**DOI:** 10.3390/bs14070570

**Published:** 2024-07-05

**Authors:** Joana Pinto, Camila Comprido, Vanessa Moreira, Marica Tina Maccarone, Carlotta Cogoni, Ricardo Faustino, Duarte Pignatelli, Nicoletta Cera

**Affiliations:** 1Faculty of Psychology and Education Sciences, University of Porto, 4200-135 Porto, Portugalup201707607@up.pt (C.C.);; 2Faculty of Medicine, University of Porto, 4200-319 Porto, Portugal; 3AUSL Pescara, “Santo Spirito” Hospital of Pescara, 65124 Pescara, Italy; maricatina.maccarone@asl.pe.it; 4Instituto de Biofísica e Engenharia Biomédica, Faculty of Sciences, University of Lisbon, 1749-016 Lisbon, Portugal; ccogoni@ciencias.ulisboa.pt; 5Research Unit in Medical Imaging and Radiotherapy, Cross I&D Lisbon Research Center, Escola Superior de Saúde da Cruz Vermelha Portuguesa, 1300-125 Lisbon, Portugal; rfaustino@esscvp.eu; 6Department of Endocrinology, Centro Hospitalar Universitário de São João, 4200-319 Porto, Portugal

**Keywords:** DMN, fMRI, human sexual behavior, systematic review, meta-analysis, naturalistic stimuli

## Abstract

The default mode network (DMN) is a complex network that plays a significant and active role during naturalistic stimulation. Previous studies that have used naturalistic stimuli, such as real-life stories or silent or sonorous films, have found that the information processing involved a complex hierarchical set of brain regions, including the DMN nodes. The DMN is not involved in low-level features and is only associated with high-level content-related incoming information. The human sexual experience involves a complex set of processes related to both external context and inner processes. Since the DMN plays an active role in the integration of naturalistic stimuli and aesthetic perception with beliefs, thoughts, and episodic autobiographical memories, we aimed at quantifying the involvement of the nodes of the DMN during visual sexual stimulation. After a systematic search in the principal electronic databases, we selected 83 fMRI studies, and an ALE meta-analysis was calculated. We performed conjunction analyses to assess differences in the DMN related to stimulus modalities, sex differences, and sexual orientation. The results show that sexual stimulation alters the topography of the DMN and highlights the DMN’s active role in the integration of sexual stimuli with sexual schemas and beliefs.

## 1. Introduction

The default mode network (DMN) is a complex brain network that encompasses several brain regions, including the posterior cingulate cortex (PCC), anterior medial prefrontal cortex (mPFC), posterior inferior parietal lobule, and several temporal regions, which are usually active during a resting state and inactive when a subject is involved in a cognitive task [1,2,3,4,5].

The DMN usually reduces its activity during the performance of a cognitive task when compared to that during a rest state. However, the DMN is not just an intrinsic system that is only actively involved in internal or self-related and stimulus-independent processing; it is also active during specific tasks.

The DMN allows for the integration of three types of processes related to the assessment of incoming sensory input: both recent and active memories, which influence the present, and a set composed of long-term memories, beliefs, and emotions that peculiarly characterize a single individual [6]. The DMN has an active and dynamic role in the modeling of external context-dependent information. Previous studies that have used naturalistic stimuli, such as real-life stories and silent or sonorous films, have found that the information processing involves a complex, hierarchical set of brain regions, including the DMN nodes [7,8]. In similar cases, the DMN regions were responsible for integrating the information that accumulates during each scenario into the stimulus in a unique way [9]. This active role is confirmed by evidence that found that the DMN is able to integrate non-perceptual aspects of aesthetic experiences, confirming a pivotal role for the MPFC [10].

According to D’Argembeau et al. [11,12], the DMN is composed of multiple interacting systems [13,14]. The PCC and the anterior mPFC (amPFC) comprise the core system, interacting with the dorsal medial prefrontal cortex (dmPFCsys) and the medial temporal lobe (MTLsys) systems. The DmPFCsys encompasses the dorsomedial prefrontal cortex and the lateral temporal cortex to the temporoparietal junction and is implicated in mentalizing, metacognition, and scene construction [15]. The MLTsys consists of the ventral mPFC (vmPFC), the posterior inferior parietal cortices, and the retrosplenial cortex. Moreover, the hippocampal and parahippocampal areas are involved in the MLTsys, which is principally implicated in prospective memory. The three systems mentioned above show a strong interaction at different levels.

The hippocampus and the parahippocampal gyri (PHGs) represent the subcortical components of the DMN. Back projections from the hippocampal system to the parietal areas are important for memory recall [16]. According to previous findings, the PHG mediates between the PCC and the hippocampus [17]. Additionally, regions of the DMN within the MTL have been observed to be relevant in episodic memory processing [18]. Nevertheless, coupling with the hippocampus and the DMN hub occurs only during episodic memory retrieval processes and not during encoding [19].

The human sexual experience involves a complex set of processes related to both external contexts and inner processes, such as thoughts, beliefs, and past experiences and memories. Several functional magnetic resonance imaging (fMRI) studies have highlighted a complex set of cortical and subcortical brain regions related to sexual arousal or specific alterations observed in pathological conditions. Sexual arousal can be conceived as a specific part of general arousal, and it is related to sexual motivation and desire [20,21,22]. Sexual desire is commonly defined as the presence of sexual thoughts, fantasies, and motivations to engage in sexual behavior in response to relevant internal and external cues and is influenced by factors such as mood, health, and attitude. This excitement status prepares the body for sexual activity and results in a state of arousal. Sexual arousal is connected to sexual desire and is defined as both subjective and physiological [20]. Most of the fMRI studies have used visual sexual stimulation designs, in which the stimuli consisted of erotic or sexually explicit videos or photos. These studies highlighted a complex set of regions involved in the complex processes and mechanisms that accomplish one of the initial stages of sexual response, such as sexual arousal. According to the classical vision from Stoléru [23], this complexity can be neurophenomenologically summarized in four components that can process the appraisal of the sexual stimulus: the motivation and the emotional, autonomic, and sexual responses. Considering this, it can be possible to conceive sexual arousal as a cycle that comes naturally. Indeed, Georgiadis and Kringelbach [24] suggested that the cycle of sexual response and sexual pleasure may be considered the center of human sexual behavior. The stages of the human sexual response can be described as “the sexual pleasure cycle”, which differentiates among distinct phases, inspired by the approach to other pleasure-seeking behaviors, like eating.

Since the DMN plays an active role in the integration of naturalistic stimuli and aesthetic perception with beliefs, thoughts, and episodic autobiographical memories, the present meta-analysis aims to quantify the involvement of the nodes of the DMN during sexual stimulation. We also aimed to identify the influence of the stimulus type on the DMN and specific alterations in the DMN’s topological configuration related to pathological conditions, sexual orientation, and transsexualism.

## 2. Materials and Methods

### 2.1. Systematic Review Protocol, Study Selection, and Study Inclusion

The present meta-analysis followed the Preferred Reporting Items for Systematic Reviews and Meta-Analyses (PRISMA) guidelines and the PICO research-question strategy protocol (Appendix A) [25]. The flowchart in Figure 1 depicts the steps of the selection process.

fMRI studies about sexual arousal evoked using visual sexual stimulation published in English between 2000 and 2024 were systematically searched in PubMed, Web of Science, and Scopus.

The computer search was based on the PICO approach, combining the keywords “sexual arousal” and “fMRI” (sexual arousal: “sexual arousal” [MeSH Terms] OR (“sexual” [All Fields] AND “arousal” [All Fields]) OR “sexual arousal” [All Fields]; fMRI: “magnetic resonance imaging” [MeSH Terms] OR (“magnetic” [All Fields] AND “resonance” [All Fields] AND “imaging” [All Fields]) OR “magnetic resonance imaging” [All Fields] OR “fmri” [All Fields]).

We selected studies that met the following criteria: (1) MNI or Talairach coordinates that were mentioned in the tables, figure captions, or results section; (2) studies that used visual, tactile, and olfactory modalities related to sexual behavior (i.e., visual or audio-visual sexual stimulation; olfactory stimulation; tactile sexual stimulation). First, the presence of tridimensional (MNI or Talairach) coordinates was checked and retrieved. Then, all the reported Talairach coordinates were transformed in MNI using the tool “convert foci” present in GingerALE. The obtained MNI coordinates were thus screened to find those representing DMN nodes according to those used by Esposito et al. [26]. We included the following bilateral AAL clusters: “CINGULUM_ANT; FRONTAL_SUP_MED; HIPPOCAMPUS; PARAHIPPOCAMPAL; TEMPORAL_MID; CINGULUM_POST; ANGULAR; PRECUNEUS; PARIETAL_INF”. Moreover, we used the AAL atlas implemented in MRIcron (v.1.0.2) to assess the correspondence between the retrieved coordinates and the DMN nodes [27].

We obtained 83 [28,29,30,31,32,33,34,35,36,37,38,39,40,41,42,43,44,45,46,47,48,49,50,51,52,53,54,55,56,57,58,59,60,61,62,63,64,65,66,67,68,69,70,71,72,73,74,75,76,77,78,79,80,81,82,83,84,85,86,87,88,89,90,91,92,93,94,95,96,97,98,99,100,101,102,103,104,105,106,107,108,109,110] studies after duplicate removal (Figure 1). Subsequently, we retrieved all the studies and screened them to identify other studies of interest. Similarly, all the narrative, systematic reviews, and meta-analyses were retrieved and checked to find relevant studies concordant with our search. Two authors independently performed the literature search and assessment (Table 1 and Figure 1).

### 2.2. ALE fMRI Meta-Analysis

We performed an activation likelihood estimation (ALE) using GingerAle 3.02 (https://www.brainmap.org/ale/ accessed on 1 January 2024) [111] on the DMN clusters reported in the included studies. Since most of the studies reported coordinates resulting from the contrast analyses, whereas some of the included studies also reported coordinates for each stimulus category or group, we decided to report both types of results.

ALE is a widely used analytical technique for performing coordinate-based meta-analyses and determining the convergence of foci, as reported in previous studies. GingerAle implements the ALE algorithm and calculates the maximum probability of activation to create modeled activation maps for each experiment. Then, the union of all modeled activation maps was computed voxel by voxel, and each sample size was taken into consideration. The obtained whole-brain ALE maps were created by comparison with a null distribution map. The reliability of an ALE map was calculated by applying permutation, which allowed for the determination of the difference between the true activation foci convergence and random clustering [112].

To assess the involvement of the DMN during sexual stimulation, an ALE map with all the DMN-related coordinates was calculated (FDR pN < 0.01 corrected). Moreover, contrast analyses were performed to assess the following:(i)Differences between videos and pictures, highlighting the DMN’s involvement in responses to dynamic and static stimuli.(ii)Similarly, differences in the DMN nodes between heterosexuals and homosexuals and between heterosexuals and transsexuals;(iii)Differences between heterosexual healthy subjects and pedophiles;(iv)Differences between heterosexual healthy men and patients affected by sexual dysfunctions, such as psychogenic erectile dysfunction or premature ejaculation.

Due to the number of studies, an FDR correction was applied when the number of studies was greater than or equal to 35. With < 35 studies, a *p* < 0.001 was applied.

Contrast analyses were carried out using conjunction analysis. Specifically, conjunction examines two different sets of foci for statistically significant differences in convergence. It contrasts and compares two datasets, showing their similarities, by calculating the voxelwise minimum value of the input ALE images.

We used Mango 4.1 (http://ric.uthscsa.edu/mango/mango.html, accessed on 1 January 2024) [113] to visualize the results by navigating between the volumes of the image of an MRI template in the MNI space with a 2 × 2 × 2 mm resolution (https://www.brainmap.org/ale/, accessed on 1 January 2024).

## 3. Results

### 3.1. Principal Characteristics of the Included Studies

The results of this meta-analysis are based on data obtained from 2587 participants, of which 1545 were men (59.72%) and 956 were women (36.95%). Among the male participants, 1220 participants identified as heterosexual (78.96%), 212 as homosexual (13.72%), and 42 as bisexual (2.72%). The sexual orientation of 71 participants was not mentioned, and they were merely reported as healthy (4.60%). Among all the male participants, 92 were patients (5.95%), 76 of whom were heterosexual (82.61%), 12 of whom were homosexual (13.04%), and 4 of whom were bisexual (4.35%). Moreover, 71 men were reported to have sexual dysfunction (4.60%), such as psychogenic erectile dysfunction (41–2.65%) and pedophilia (30–1.94%). Among those in the pedophilia group, 14 were heterosexual (46.67%), 12 were homosexual (40.00%), and 4 were bisexual (13.33%). Another 10 men were reported to have major depression disorder (MDD—0.64%), and 11 had infarctions in the right middle cerebral artery (MCA) territory (0.71%).

Of all the female participants, 671 were heterosexual (70.19%), 90 were homosexual (9.41%), 24 were bisexual (2.51%), and 171 were healthy individuals (17.89%). Among the women, only 27 (2.82%) were breast cancer survivors.

This meta-analysis included not only cisgender participants but also transsexual and nonbinary participants. More specifically, 45 of the participants were transsexual male-to-female (1.74%), 30 were transsexual female-to-male (1.16%), and 11 were nonbinary (0.43%).

The age range of the participants was between 18 and 56 years, with an average of 30.66 years. The mean age was 31.63 years for the male participants, 25.48 for the female participants, 36 for the transsexual participants, and 37 for the nonbinary participants.

Most of the studies were from Germany (27.71%), South Korea (19.28%), and the United States of America (16.87%). The rest were from the Netherlands (6.02%), Italy (6.02%), Canada (4.82%), China (4.82%), the United Kingdom (4.82%), Switzerland (3.61%), France (2.41%), Spain (1.20%), Sweden (1.20%), and Austria (1.20%).

Most studies used only pictures (54.22%) or videos (37.35%) as stimuli. Only two studies used both pictures and videos (2.41%). Moreover, two studies used pictures in combination with a task, one with an approach-avoidance task (1.20%), and the other with a choice reaction time task (1.20%). The rest used video games (1.20%), self-induced orgasm (1.20%), or genital (0.97%) or olfactory stimulation (1.20%).

There was some variation in terms of the magnetic field strength of the scanners used in the studies. The majority of the studies used 3T MRI scanners (59.04%). The remaining studies used 1.5 T (37.35%), 7 T (2.41%), and 2 T (1.20%) MRI scanners. A total of 62 studies performed whole-brain analysis (74.70%), 16 performed ROI-based analysis (19.28%), and 5 performed a combination of both (6.02%).

In four of the studies, the effects of medication were examined (4.82%). Of these, three had randomized double-blind designs (75.00%), two were placebo-controlled (50.00%), one was within-subjects (25.00%), one had a crossover design (25.00%), and one compared only two groups of women who were taking or not taking oral contraception during their menstrual cycle (25.00%). All the studies assessed different medications, such as bupropion and paroxetine, gamma-hydroxybutyrate (GHB), oral contraception, and kisspeptin.

Furthermore, of all 83 studies, 60 used MNI coordinates (72.29%), and 23 used Talairach coordinates (27.71%; Figure 2).

### 3.2. Brain Results

To assess the involvement of the DMN during sexual stimulation in healthy heterosexual participants or in specific populations or pathological conditions, we performed a series of meta-analyses, followed by conjunction analyses, which allowed for a comparison of the results. During sexual stimulation, all the principal nodes included in the DMN were involved. However, more homogeneous clusters, with higher ALE values, were found in correspondence with the ACC/mPFC and parahippocampal/hippocampi bilaterally (FDR pN < 0.01; Figure 3).

As mentioned above, most of the studies applied visual sexual stimulation, using silent, audiovisual stimuli or pictures/photos. To assess topological alterations in the involvement of DMN nodes during sexual stimulation, we performed a conjunction analysis to compare the use of video and pictures. This analysis highlighted the involvement of subcortical components of the DMN (FDR pN < 0.01) during video > pictures in correspondence with the bilateral parahippocampal and hippocampal regions.

To study the presence of between-sex differences in the DMN, we compared the studies that investigated sexual arousal in heterosexual men and women (men ∩ women) using a conjunction analysis, with uncorrected *p* < 0.001, since 40 studies for men and 13 for women were included. The results highlight convergent significant clusters corresponding to the ACC/mPFC and parahippocampal/hippocampi bilaterally and to the left angular gyrus/IPL (Table 2).

Similarly, the contrast between heterosexuals (both males and females) and (∩) homosexuals (both males and females) showed homogenous convergent clusters in the left parahippocampus and dorsomedial PFC/ACC (Table 3; Figure 4).

The studies that investigated patients with mood disorders, sexual dysfunctions, MTF and FTM transsexuals, and pharmacological trials were not included in the contrast analyses due to the low number of studies (n < 10) and fewer foci.

## 4. Discussion

Despite the growing interest in the cerebral underpinnings of human sexual behavior and sexual arousal, little is known about the role of the DMN in the processing of sexual stimuli. Most of the information about the role played by the DMN in sexual behavior is from clinical fMRI studies. In recent years, several fMRI studies have investigated alterations in the DMN nodes during the resting state, predominantly involving male participants. According to our systematic search, only 83 of the 103 studies showed specific results and coordinates corresponding to the DMN nodes and hubs. The global ALE map showed that during sexual stimulation in all the studies, the topography of the DMN was preserved, including all the principal cortical and subcortical nodes. Naturalistic, emotionally arousing stimuli allow for the activation of the DMN. Several studies have shown that the DMN plays an active role during the processing of audiovisual stimuli, such as silent movies and listening to stories [114]. Interestingly, our results reveal wider involvement and more convergent results in the large anterior cluster, which brings together the ACC and mPFC. The ACC and mPFC together represent one of the hubs of the DMN. Conversely, we found weaker results corresponding to the hubs, including the PCC and precuneus. This result can confirm a specific role played by the anterior cingulate and medial prefrontal cortices during the elaboration of complex naturalistic stimuli and with the formation and recall of schemas. The formation and recall of specific schemas rely on regions such as the ventromedial prefrontal cortex, precuneus, bilateral temporoparietal junction, and hippocampus [115]. Sexual intercourse, as shown in a video, involves a series of schemas. For instance, several studies have shown that all video stimuli show specific phases of sexual intercourse, such as petting and vaginal or oral intercourse. Despite this, sexual self-schema is a part of a broader concept of the self that is believed to be crucial for intrapersonal and interpersonal sexual relationships [116]. Previous findings have revealed that positive schematic women and men reported higher levels of sexual self-efficacy and lower levels of sexual aversion [117,118].

We found that visual stimuli, such as pictures and videos, were the main modality used to elicit genital or subjective sexual arousal. Among the brain networks, the DMN was found to be relevant for carrying an internal model of information from narrative content, and its clusters were found to play an important role in the representation of event models and schemas [119,120]. The response of the DMN was hypothesized to be strictly related to the content of the stimuli over long time scales and is invariant to changes in low-level properties of the stimuli [6]. Moreover, studies that used the same content presented with different modalities (i.e., audiovisual vs. written content) found no dramatic changes in the temporal DMN responses [119,121]. However, according to our findings, different stimulus modalities and paradigms elicited different DMN responses. Comparisons between dynamic and static stimuli, such as videos and pictures, respectively, highlighted the role played by subcortical nodes of the DMN. Convergent results were found in the bilateral parahippocampal regions. Coactivation of the DMN and subcortical regions, such as the hippocampus, was found during naturalistic movie viewing and was related to the fluctuation in surprise experienced by the participants [114]. During surprise, the hippocampal regions included in the DMN integrate three different processes relevant to the experience of surprise: switching between unexpected external information in input, episodic memories related to incoming information, and the internal model. However, the angular gyrus plays a relevant role in this integration process [122]. No involvement of the angular gyrus was observed for the comparison between videos and pictures in our study. The involvement of the parahippocampus could also be related to the sensitivity of the DMN to long-term memories. Sexual activities, as depicted in a video, could induce the retrieval of long-term content-related memories. In a recent study, an excerpt from the *Twilight Zone* was shown to participants. Those who had a background in the plot of the TV series showed increased connectivity between the DMN nodes and the hippocampus [123].

Most of the studies have investigated the brain correlates of sexual arousal in healthy heterosexual male participants, representing a great methodological and theoretical limitation that can indirectly affect our results and their interpretation [124]. We observed significant sex differences in the DMN, which partially preserved its topography with a pattern involving the mPFC, parahippocampus, and left precuneus/IPL.

The DMN is highly sensitive to individual differences in the interpretation of external incoming content. Most of the sexual stimuli used in previous studies were adapted to a male heterosexual audience. For instance, it is possible to hypothesize that women inexperienced or unfamiliarized with explicit sexual video clips might feel more discomfort or be more negatively affected than when viewing erotic or romantic ones. In this way, Borg et al. [31] demonstrated a similar brain activation pattern to “explicit penetrative” or sadomasochistic and disgusting stimuli in women. Moreover, only a few studies have taken into account hormonal cycling shifts that could affect desire, sexual motivation, and arousal in women. However, as mentioned above, the active DMN integrates incoming information over a long time scale with schemas, beliefs, and memories. Beliefs play a crucial triggering role in sexual arousal in men. Similarly, sexual and erotic fantasies can be a trigger for sexual arousal in women. However, our results seem to confirm that male sexual arousal is more stimulus-dependent than fantasy-evoked [125].

In recent years, some studies have shown that kisspeptin could play a role in sexual motivation and desire in both men and women [126,127]. Kisspeptin administration can negatively affect the role of the dmPFC system in sexual processing in healthy men. This neuropeptide can improve desire and drive aspects of motivation [128]. In a randomized clinical trial, Thurston and colleagues [128] reported that the administration of kisspeptin to premenopausal women with hypoactive sexual desire disorder improved their brain response to sexual stimuli and attractive faces. Interestingly, they found an association between kisspeptin and activation of the hippocampus during visual sexual stimulation. The hippocampus and middle and inferior frontal regions contain a high density of kisspeptin receptor-1 (KISS1R) [129].

Similar to the contrast observed between men and women, the contrast between heterosexual and homosexual participants highlights the interplay between the mPFC/ACC node and the left parahippocampus. A recent study indicated that 98% of homosexual men reported having viewed pornography within 30 days, which is more than the 72% of heterosexual men [130]. Our results not only show an alteration in the DMN pattern but also confirm a possible role of its previously mentioned nodes in the integration between schemas and incoming information processing. However, in homosexual women and men, sexual self-schemas have been less studied, and less is known about their influence on contextual incoming information processing. Moreover, Lau et al. [131,132] reported that 42.5% of gay men and 75.6% of lesbian women reported at least one sexual problem during their lifetime. These sexual problems can be reflected in negative sexual schemas that are common among homosexuals with mood disorders. Compared to heterosexual individuals, homosexual individuals tend to activate cognitive schemas of incompetence, self-depreciation, and helplessness [133].

## 5. Conclusions and Limitations

The present ALE meta-analysis shows, for the first time, a possible active role of the DMN during the processing of sexual stimuli. Moreover, our results highlight topographic changes in men, women, and homosexuals that could underlie differences not only in the processing of incoming information but also in its integration with schemas, beliefs, and memories that play a crucial role in human sexual behavior. Intriguingly, the results shown in this study highlight the importance of sexual stimulation as a tool to discover specific alterations in the integration between incoming information schemas and beliefs. This could provide several insights that are helpful for developing new psychotherapy strategies to treat sexual dysfunctions and sexual perturbation. Following new advances, the role played by other brain networks also needs to be assessed [134]. Further studies could assess the intersubject connectivity during sexual stimulation [135] and the relevance of the DMN during ecological experiences using different imaging techniques, such as functional near-infrared spectroscopy (fNIRS) [136].

However, several limitations need to be acknowledged. First, as we stated, we were not able to perform specific analyses to assess alterations in the DMN activity during sexual stimulation in transsexuals, pedophiles, or patients. Furthermore, our results did not consider the differences related to the experimental paradigms used. Similarly, some studies have indicated the presence of healthy participants without mentioning sexual orientation.

## Figures and Tables

**Figure 1 behavsci-14-00570-f001:**
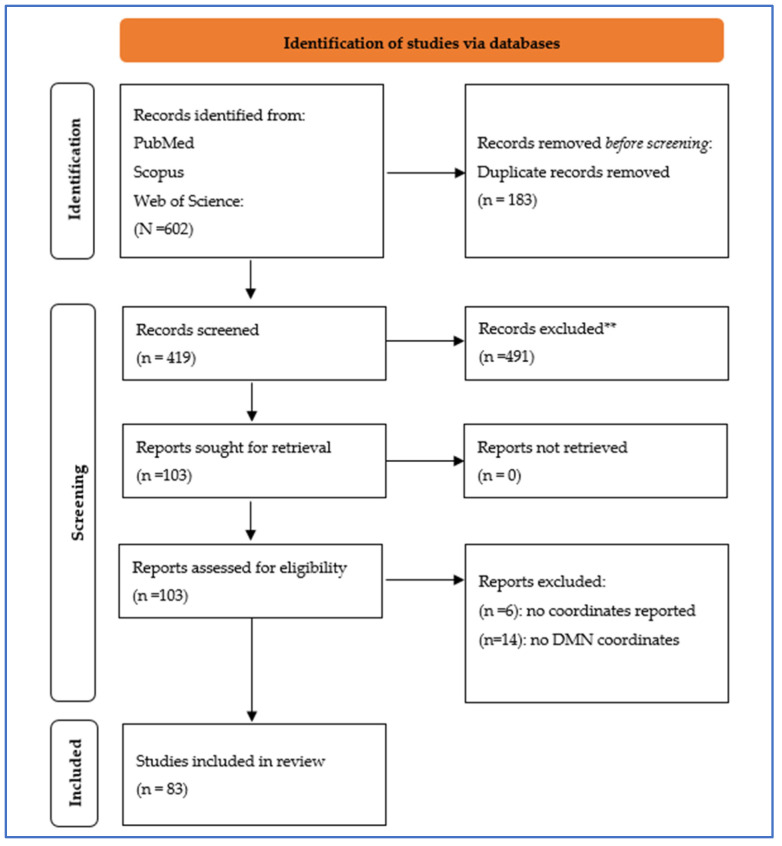
Flowchart of the selection process. ** From Page MJ, et al. [25].

**Figure 2 behavsci-14-00570-f002:**
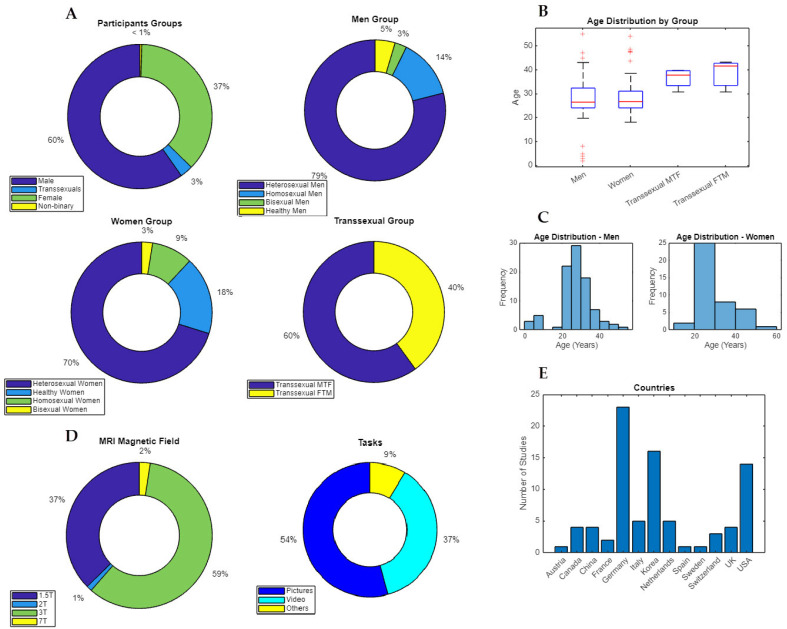
Characteristics of the included studies. (**A**). Four doughnut charts of the number of participants for each gender group (men, women, transsexuals, and nonbinary) and each sexual orientation subgroup for men and women (heterosexual, homosexual, bisexual, or healthy) and for transsexuals (transsexual male to female and transsexual female to male) (**B**). Boxplot of the age distribution of men, women, and both transsexual subgroups (**C**). Histograms of the age distribution of the male and female participants (**D**). Doughnut charts related to the number of types of MRI machine magnetic field strengths and tasks used in the studies (**E**). Bar plot of the number of studies produced by each country. All the graphs were created using MATLAB (vers. 2022b).

**Figure 3 behavsci-14-00570-f003:**
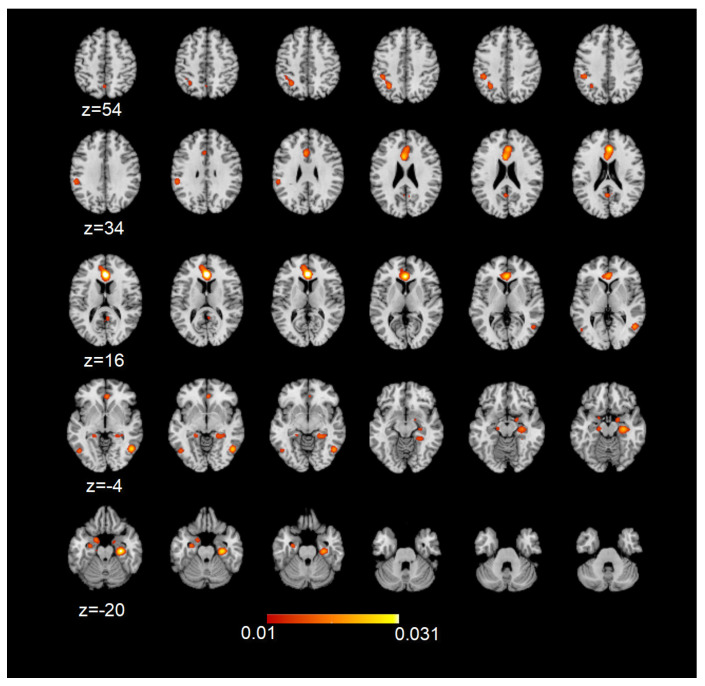
ALE map of the resulting DMN nodes during sexual stimulation. Maps are overimposed on a 2 × 2 × 2 mm MNI template according to neurological convention. The colored bar denotes the corresponding ALE value ranges indicated on the maps (FDR pN < 0.01).

**Figure 4 behavsci-14-00570-f004:**
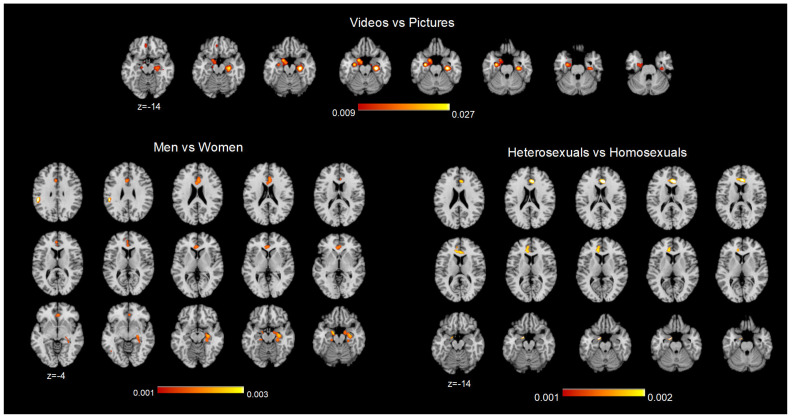
Conjunction ALE map results. Activation likelihood maps resulting from the contrast between videos and pictures (TOP), between men and women (bottom left), and between heterosexual and homosexual subjects (bottom right). The contrast map “video vs. pictures” (FDR pN < 0.01) showed convergent activity in the ventral parahippocampal regions of the DMN. The two contrast maps at the bottom (*p* < 0.001) show the involvement of both subcortical and cortical DMN nodes in sex differences and sexual preference. The maps are superimposed on an MNI 2 × 2 × 2 mm template according to neurological convention. The colored bars under each map indicate the ALE value ranges.

**Table 1 behavsci-14-00570-t001:** Inclusion and exclusion criteria for the study selection.

Inclusion Criteria	Exclusion Criteria
Design	
Experimental studies	Other designs Systematic reviews/meta-analyses
Population	
Men Women	Animals Children
Intervention	
fMRI Sexual stimulation VSS (videos, pictures)	EEG MEG fNIRS
Topic	
Sexual behavior Brain activity Default Mode Network Sexual dysfunctions Paraphilia Sexual offenders Sexual orientation Transsexualism	Other brain networks, different from DMN No sex-related studies

**Table 2 behavsci-14-00570-t002:** Brain clusters resulting from ALE meta-analysis performed for all the studies.

Cluster	BA	Hemisphere	*x*	*y*	*z*	ALE	*p*	Z
Anterior cingulate cortex	24	L	0	34	14	0.04148	0.00000	9.389
Middle temporal gyrus	21	R	50	−60	0	0.02803	0.00000	7.215
Posterior cingulate cortex	23	L	−2	−54	22	0.01799	0.00000	5.306
Parahippocampal gyrus	--	L	−30	−8	−22	0.01898	0.00000	5.510
Parahippocampal gyrus	--	L	−22	−36	−4	0.01617	0.00000	4.917
Posterior cingulate cortex	31	R	6	−56	26	0.01382	0.00001	4.390

Abbreviations: BA = Brodmann area; L = left; R = right.

**Table 3 behavsci-14-00570-t003:** Brain clusters resulting from the conjunction analysis between video and picture stimuli, men and women, and heterosexual and homosexual participants (*p* < 0.001).

Cluster	BA	Hemisphere	*x*	*y*	*z*	ALE
**Videos vs. Pictures (FDR pN < 0.01)**						
Parahippocampal gyrus	28	R	22	−14	−20	0.01400
Parahippocampal gyrus	34	L	−17	−2	−22	0.01210
**Men vs. Women (*p* < 0.001).**						
Anterior cingulate cortex	32	L	−4	42	14	0.00202
Medial frontal gyrus	9	L	−4	50	8	0.00192
Parahippocampal gyrus	28	L	−18	−4	−20	0.00335
Parahippocampal gyrus	28	R	34	−18	−28	0.00190
Precuneus		L	−2	−60	50	0.00184
**Heterosexuals vs. Homosexuals (*p* < 0.001).**						
Anterior cingulate cortex	32	L	−12	38	10	0.00230
Anterior cingulate cortex	32	L	−12	34	12	0.00228
Inferior parietal lobule	40	L	−32	−52	52	0.00229
Parahippocampal gyrus	34	L	−18	0	−22	0.00325
Anterior cingulate cortex	32	R	4	42	−2	0.00192

Abbreviations: BA = Brodmann area; L = left; R= right.

## Data Availability

Not applicable.

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
