# Peer review of "The Complex Role Played by the Default Mode Network during Sexual Stimulation: A Cluster-Based fMRI Meta-Analysis"

_behavsci, 2024, doi:10.3390/bs14070570_

Round 1

Reviewer 1 Report

Comments and Suggestions for Authors

The manuscript is a review and meta-analysis investigating whether the Default Mode Network (DMN) can code the brain processes underlying human sexuality and its alterations. The study integrates findings from eleven studies using Activation Likelihood Estimation (ALE) fMRI meta-analysis. It concludes that DMN and its subsystems play a significant role in social cognition and autobiographical memory processes related to human sexual behavior.

While the study has some merits, it has profound limitations and methodological biases which should be solved. Authors should re-run the literature search and analyses, re-analyze data, report their pipeline and results more accurately and comprehensively, and make sure that (1) their study is reproducible and (2) their claims are supported by the literature/the findings.

I. Selection Bias.

The study incorporates a selection bias which needs to be overcome. The study appears to incorporate a selection bias by focusing the search strategy exclusively on studies that explicitly mention the DMN in relation to human sexual behavior. This narrow scope likely excludes relevant studies investigating the neural correlates of sexual behavior without directly referencing to DMN, potentially skewing results towards those confirming pre-existing hypotheses and undermining the ability to draw unbiased conclusions about the DMN's specificity and role. To deepen the understanding of DMN's contribution to sexual behavior, it is crucial to adopt a more comprehensive search strategy that includes a broader spectrum of studies. Additionally, employing sensitivity analyses and network connectivity analyses can reveal the interaction between DMN and other brain networks, thereby enhancing the robustness and transparency of the findings. This approach will allow for a more accurate depiction of the DMN's role within the broader neurobiological context of human general behavior and specific sexual behavior, beyond the confines of direct references in the literature.

II. Data Extraction and Manipulation

The method by which data were extracted and synthesized is crucial, especially in neuroimaging studies where different studies might use varying fMRI protocols and analysis techniques. Please be sure to accurately specify how data was extracted, how was it handled, and how analyses were performed including analysis details.

·         The manuscript uses Activation Likelihood Estimation, which is appropriate for the data type but depends heavily on the accurate reporting of coordinates and the quality of individual studies' fMRI data. Please explicitly state how study coordinates were standardized and synthesized across studies. Also, clarify how the ALE scores were calculated and interpreted since the section 2.3 is unexpectedly short in details.

·         How did the authors address the variability in fMRI data acquisition and analysis across studies?

·         Did the author assess whether the included studies were sufficiently homogeneous in terms of populations studied, specific sexual behaviors examined, and imaging techniques used? Heterogeneity could complicate the interpretation of meta-analysis results.

·         Overall, the authors should check that their analysis is fully reproducible before resubmitting their manuscript.

III. Theoretical Issues.

·         The manuscript posits several connections between the Default Mode Network (DMN) and human sexual behavior, extending some claims into areas with less empirical support, specifically concerning DMN's direct involvement in sexual processes. While it's plausible to consider DMN and its subsystems in the context of sociocognitive processes underlying human sexuality, such claims necessitate cautious interpretation. The primary functions associated with DMN—such as mind-wandering, self-referential thought, and memory—are typically activated in a wide range of psychological processes, not exclusively sexual behavior. This suggests that the interpretation that DMN supports sexual behavior could inadvertently conflate the network's general role in self-directed consciousness with specific sexual cognitive processes. Indeed, the observed DMN activation might be a by-product of its role in self-directed consciousness, which is likely engaged during the erotic stimuli used in the 11 studies analyzed. Such a scenario underscores the potential for overinterpretation where DMN's activation is viewed as directly related to sexual behavior, rather than being an artifact of the experimental paradigms employed. More accurate experimental designs or analytical methods are needed to distinguish between DMN's general cognitive functions and its specific contributions to sexual behavior, potentially through comparative analyses involving different types of cognitive tasks.

·         As with any correlational research, especially in neuroimaging, caution must be exercised in describing causa associations. Can the results genuinely support causal inferences, or are they merely associations? Are these conclusions generalizable across different populations, sexual orientations, and types of sexual behavior?

·         The authors seem to rely on an obsolete conception of DMN as a "default" network (e.g., wakeful rest, line 43), which is wrong. Moreover, the authors seem to rely on semi-obsolete characterisations of the brain architecture. Please consider integrating more knowledge on the topics (Spreng, 2012; Glasser et al., 2016; Di Plinio and Ebisch, 2018; Brandman et al., 2020; Yeshurun et al., 2021; Jeong and Paolini, 2023).

·         Please discuss the above criticalities considering that scientific research and interpretations should be objective and unbiased.

“Minor”

·         Please change the title. Brain networks are not tools.

·         What do the authors mean with "alterations of human sexuality"?

·         In line 149-150, the sentence “Brain imaging studies about sexual dysfunctions, perturbations, or sexual orientation, in both women and men, and DMN were included (P-Population)” appears to be nonsensical.

·         The label of Figure 3 appears to be wrong.

·         Moreover, Figure 3 reports results ambiguously: Please indicate error bars; Please indicate each process with the same color; Please explain accurately how the brain maps were obtained; Please explain the meaning of the denser red colors in the brain maps.

·         Lines 354-355: To investigate the role of a network, subnetwork, or subsystem, one could not focus ONLY on such set of regions. In fact, the "role" of a brain component automatically also encompasses the specificity of its contribution to behavior or behavioral variability. By focusing only on DMN-related studies, authors are not able to draw any conclusion about the specificity of the DMN.

References

Brandman, T., Malach, R., & Simony, E. (2020). The surprising role of the default mode network in naturalistic perception. Communications Biology, 4. https://doi.org/10.1038/s42003-020-01602-z.

Di Plinio, S., Ebisch, S. J. H. (2018). Brain Network Profiling defines functionally specialized cortical networks. Human Brain Mapping, 39(12): 4689–4706. doi: 10.1002/hbm.24315.

Glasser, M. F., Coalson, T. S., Robinson, E. C., Hacker, C. D., Harwell, J., Yacoub, E., … van Essen, D. C. (2016). A multi-modal parcellation of human cerebral cortex. Nature, 536(7615), 171–178

Jeong, G., & Paolini, M. (2023). Dynamic Adaptation of Default Mode Network in Resting state and Autobiographical Episodic Memory Retrieval State. bioRxiv. https://doi.org/10.1101/2022.08.30.505423.

Spreng, R. N. (2012). The fallacy of a “task-negative” network. Frontiers in Psychology, 3(145), 1–5. https://doi.org/10.3389/fpsyg.2012.00145

Yeshurun, Y., Nguyen, M., & Hasson, U. (2021). The default mode network: where the idiosyncratic self meets the shared social world. Nature Reviews Neuroscience, 22, 181-192. https://doi.org/10.1038/s41583-020-00420-w.

Comments on the Quality of English Language

Some English expressions are ambiguous. I indicated some of these in my comments but please check the whole manuscript for English and transparency.

Author Response

Reviewer 1

The manuscript is a review and meta-analysis investigating whether the Default Mode Network (DMN) can code the brain processes underlying human sexuality and its alterations. The study integrates findings from eleven studies using Activation Likelihood Estimation (ALE) fMRI meta-analysis. It concludes that DMN and its subsystems play a significant role in social cognition and autobiographical memory processes related to human sexual behavior.

While the study has some merits, it has profound limitations and methodological biases which should be solved.

Authors: Thank you very much for your insightful suggestions and recommendations that helped to improve our study.

Authors should re-run the literature search and analyses, re-analyze data, report their pipeline and results more accurately and comprehensively, and make sure that (1) their study is reproducible and (2) their claims are supported by the literature/the findings.

Authors: Thanks. We agree with you. For this reason, we have performed a new literature search as you can read in the methods (figure 1) and the introduction, in the paragraph about the hypotheses. Indeed, following your suggestions, we performed a new literature search based on the generic terms that are, however, reproducible, as you can read in the methods. Moreover, as we described, we searched the coordinates of the principal nodes of DMN, following previous studies and in particular Esposito et al., 2018 (a resting state study about the correlation between DAN and DMN). To check the DMN coordinates, we used mricron that implemented the AAL atlas MNI based. This is to be sure that all the MNI (transformed in the case of Talairach coordinates with ginger ALE) related to DMN were included in the specific regions (as reported in the main text).    

  1. Selection Bias.

The study incorporates a selection bias which needs to be overcome. The study appears to incorporate a selection bias by focusing the search strategy exclusively on studies that explicitly mention the DMN in relation to human sexual behavior. This narrow scope likely excludes relevant studies investigating the neural correlates of sexual behavior without directly referencing to DMN, potentially skewing results towards those confirming pre-existing hypotheses and undermining the ability to draw unbiased conclusions about the DMN's specificity and role. To deepen the understanding of DMN's contribution to sexual behavior, it is crucial to adopt a more comprehensive search strategy that includes a broader spectrum of studies. Additionally, employing sensitivity analyses and network connectivity analyses can reveal the interaction between DMN and other brain networks, thereby enhancing the robustness and transparency of the findings.

This approach will allow for a more accurate depiction of the DMN's role within the broader neurobiological context of human general behavior and specific sexual behavior, beyond the confines of direct references in the literature.

Authors: Thank you. Indeed, we agree with you, but the reason that has channeled our interest in investigating the role played by DMN in sexual behavior was similar to that expressed in your appraisal and presented in a previous paper. (i.e. Cera et al 2014, Plos One; the first study that investigated the role of intrinsic brain network during long visual erotic stimulation in psychogenic ED and healthy men; Following the line from Pamilo et al., 2011). However, we decided to orient our attention to the DMN and we found and selected only a few studies that applied different analytical techniques on different clinical/non-clinical populations. This may reasonably be the ancient classical "the egg and the hen dilemma" that could be translated as “the search-strategy and heterogeneity dilemma”, that is the restriction of the strategy search terms to “DMN+Sexual Behavior+fMRI” resulting in heterogeneous results.  Following your suggestion, we searched all the studies that investigated brain underpinnings of sexual arousal with different types of stimuli and in different populations. However, we hope that this new analysis, as well as the new approach, was able to overcome the selection bias that you mentioned and that affected our previous meta-analysis.

However, we chose to carry out an ALE meta-analysis, instead of a sensitivity analysis with SDM, and as we explain in the main text (please, see the methods and results), a series of conjunction analyses to study the topographical DMN following the stimuli used in the included its ( i.e. videos vs pictures), as well as sex differences (i.e. men vs women), and sexual orientation (heterosexuals vs homosexuals). Due to the scarcity of studies (n<10) that investigated brain correlates of sexual behavior in transsexuals, patients with sexual dysfunctions, psychiatric patients, and pedophiles, we avoid calculating single dataset meta-analyses, and conjunctions.

  1. Data Extraction and Manipulation

The method by which data were extracted and synthesized is crucial, especially in neuroimaging studies where different studies might use varying fMRI protocols and analysis techniques. Please be sure to accurately specify how data was extracted, how was it handled, and how analyses were performed including analysis details.

  • The manuscript uses Activation Likelihood Estimation, which is appropriate for the data type but depends heavily on the accurate reporting of coordinates and the quality of individual studies' fMRI data. Please explicitly state how study coordinates were standardized and synthesized across studies. Also, clarify how the ALE scores were calculated and interpreted since the section 2.3 is unexpectedly short in details.

Authors: Thank you. As you requested, we improved the method section, describing all the phases of the analyses. Please check the methods section.

  • How did the authors address the variability in fMRI data acquisition and analysis across studies?
  • Did the author assess whether the included studies were sufficiently homogeneous in terms of populations studied, specific sexual behaviors examined, and imaging techniques used? Heterogeneity could complicate the interpretation of meta-analysis results.
  • Overall, the authors should check that their analysis is fully reproducible before resubmitting their manuscript.

Authors: Thank you. As we previously stated we have restructured all the manuscript.

III. Theoretical Issues.

  • The manuscript posits several connections between the Default Mode Network (DMN) and human sexual behavior, extending some claims into areas with less empirical support, specifically concerning DMN's direct involvement in sexual processes. While it's plausible to consider DMN and its subsystems in the context of sociocognitive processes underlying human sexuality, such claims necessitate cautious interpretation. The primary functions associated with DMN—such as mind-wandering, self-referential thought, and memory—are typically activated in a wide range of psychological processes, not exclusively sexual behavior. This suggests that the interpretation that DMN supports sexual behavior could inadvertently conflate the network's general role in self-directed consciousness with specific sexual cognitive processes. Indeed, the observed DMN activation might be a by-product of its role in self-directed consciousness, which is likely engaged during the erotic stimuli used in the 11 studies analyzed. Such a scenario underscores the potential for overinterpretation where DMN's activation is viewed as directly related to sexual behavior, rather than being an artifact of the experimental paradigms employed. More accurate experimental designs or analytical methods are needed to distinguish between DMN's general cognitive functions and its specific contributions to sexual behavior, potentially through comparative analyses involving different types of cognitive tasks.

Authors: Thank you. As you suggested, we have changed our theoretical plan about DMN and sexual behavior. Following your suggestions, we have restructured the introduction highlighting the active role played by DMN in naturalistic stimulation.

  • As with any correlational research, especially in neuroimaging, caution must be exercised in describing causa associations. Can the results genuinely support causal inferences, or are they merely associations? Are these conclusions generalizable across different populations, sexual orientations, and types of sexual behavior?
  • The authors seem to rely on an obsolete conception of DMN as a "default" network (e.g., wakeful rest, line 43), which is wrong. Moreover, the authors seem to rely on semi-obsolete characterisations of the brain architecture. Please consider integrating more knowledge on the topics (Spreng, 2012; Glasser et al., 2016; Di Plinio and Ebisch, 2018; Brandman et al., 2020; Yeshurun et al., 2021; Jeong and Paolini, 2023).

Authors: Thank you. Following your suggestion, we have revised the concept of DMN  more related to the resting state than to be related to naturalistic stimulation, and thus sexual stimulation.

  • Please discuss the above criticalities considering that scientific research and interpretations should be objective and unbiased.

“Minor”

  • Please change the title. Brain networks are not tools.
  • What do the authors mean with "alterations of human sexuality"?
  • In line 149-150, the sentence “Brain imaging studies about sexual dysfunctions, perturbations, or sexual orientation, in both women and men, and DMN were included (P-Population)” appears to be nonsensical.
  • The label of Figure 3 appears to be wrong.
  • Moreover, Figure 3 reports results ambiguously: Please indicate error bars; Please indicate each process with the same color; Please explain accurately how the brain maps were obtained; Please explain the meaning of the denser red colors in the brain maps.
  • Lines 354-355: To investigate the role of a network, subnetwork, or subsystem, one could not focus ONLY on such set of regions. In fact, the "role" of a brain component automatically also encompasses the specificity of its contribution to behavior or behavioral variability. By focusing only on DMN-related studies, authors are not able to draw any conclusion about the specificity of the DMN.

 Authors: Thank you. As you can read, the manuscript was completely rewritten, but we appreciate your feedback.

References

Brandman, T., Malach, R., & Simony, E. (2020). The surprising role of the default mode network in naturalistic perception. Communications Biology, 4. https://doi.org/10.1038/s42003-020-01602-z.

Di Plinio, S., Ebisch, S. J. H. (2018). Brain Network Profiling defines functionally specialized cortical networks. Human Brain Mapping, 39(12): 4689–4706. doi: 10.1002/hbm.24315.

Glasser, M. F., Coalson, T. S., Robinson, E. C., Hacker, C. D., Harwell, J., Yacoub, E., … van Essen, D. C. (2016). A multi-modal parcellation of human cerebral cortex. Nature, 536(7615), 171–178

Jeong, G., & Paolini, M. (2023). Dynamic Adaptation of Default Mode Network in Resting state and Autobiographical Episodic Memory Retrieval State. bioRxiv. https://doi.org/10.1101/2022.08.30.505423.

Spreng, R. N. (2012). The fallacy of a “task-negative” network. Frontiers in Psychology, 3(145), 1–5. https://doi.org/10.3389/fpsyg.2012.00145

Yeshurun, Y., Nguyen, M., & Hasson, U. (2021). The default mode network: where the idiosyncratic self meets the shared social world. Nature Reviews Neuroscience, 22, 181-192. https://doi.org/10.1038/s41583-020-00420-w.

Authors: Thank you. As you can read in the manuscript, and the reference list, we have greatly appreciated your suggestions.

REVIEWER 2

This article presents a topic of interest, but there are some points for a major revision:

The introduction includes some sentences (at the beginning) which are vague. Please delete and present a more focused introduction.

Authors: Thank you very much for your appraisal and recommendations that helped to improve our study. However, since the Reviewer #1 asked for an extensive revision that implied carrying out a new cluster-based meta-analysis. You can read our reply to Reviewer #1’s commentaries and suggestions and you can read the new version of our study. 

Table 3 is difficult to follow.

Authors: Thanks. We have removed it and now you can read the new results section. 

3.2. is not justified and authors have to explain the selection of these tests to the readers.

Was there an age cutoff for the research papers included? For example, different findings have been found for different age groups and sexuality.

Authors: Thank you. We did not include an age cut-off, but we performed contrast analyses of specific sex differences and differences between homosexuals and heterosexuals in the DMN activity pattern. 

At the end, authors can discuss these age differencs in sexuality and related social problems in a more critical way their findings (https://www.igi-global.com/chapter/elder-abuse-and-consent-capacity/197829).

Authors: Thank you. Unfortunately, as you can read in the results, we found only adults, and we were not able to calculate age –related differences in DMN.

In addition, it is not clear whether sex is taken into consideration in the article search (please read and discuss https://www.tandfonline.com/doi/full/10.1080/14681990500238802 as well as more recent research on this topic). Authors may need to restructure the presentation of their findings as well as rerun their research according to sex/gender and age groups

Authors: Thank you. Since the whole manuscript has been restructured, we also have taken into consideration sex differences, which were discussed in the specific section.

Reviewer 2 Report

Comments and Suggestions for Authors

This article presents a topic of interest, but there are some points for a major revision:

The introduction includes some sentences (at the beginning) which are vague. Please delete and present a more focused introduction.

Table 3 is difficult to follow.

3.2. is not justified and authors have to explain the selection of these tests to the readers.

Was there an age cutoff for the research papers included? For example, different findings have been found for different age groups and sexuality.

At the end, authors can discuss these age differencs in sexuality and related social problems in a more critical way their findings (https://www.igi-global.com/chapter/elder-abuse-and-consent-capacity/197829). In addition, it is not clear whether sex is taken into consideration in the article search (please read and discuss https://www.tandfonline.com/doi/full/10.1080/14681990500238802 as well as more recent research on this topic). Authors may need to restructure the presentation of their findings as well as rerun their research according to sex/gender and age groups.

Comments on the Quality of English Language

Μinor English language editing.

Author Response

(The authors gave the same response as above.)
